# Aggregation of an Amyloidogenic Peptide on Gold Surfaces

**DOI:** 10.3390/biom13081261

**Published:** 2023-08-18

**Authors:** David L. Cheung

**Affiliations:** School of Biological and Chemical Sciences, University of Galway, H91 TK33 Galway, Ireland; david.cheung@universityofgalway.ie

**Keywords:** peptide aggregation, molecular dynamics simulation, protein adsorption

## Abstract

Solid surfaces have been shown to affect the aggregation and assembly of many biomolecular systems. One important example is the formation of protein fibrils, which can occur on a range of biological and synthetic surfaces. The rate of fibrillation depends on both the protein structure and the surface chemistry, with the different molecular and oligomer structures adopted by proteins on surfaces likely to be crucial. In this paper, the aggregation of the model amyloidogenic peptide, Aβ(16–22), corresponding to a hydrophobic segment of the amyloid beta protein on a gold surface is studied using molecular dynamics simulation. Previous simulations of this peptide on gold surfaces have shown that it adopts conformations on surfaces that are quite different from those in bulk solution. These simulations show that this then leads to significant differences in the oligomer structures formed in solution and on gold surfaces. In particular, oligomers formed on the surface are low in beta-strands so are unlike the structures formed in bulk solution. When oligomers formed in solution adsorb onto gold surfaces they can then restructure themselves. This can then help explain the inhibition of Aβ(16–22) fibrillation by gold surfaces and nanoparticles seen experimentally.

## 1. Introduction

Since their discovery amyloid fibrils have attracted significant scientific interest [1]. Much of this has focused on their involvement in disease, initially inspired by the discovery of plaques comprised of amyloid-beta fibrils in the brains of patients with Alzheimer’s disease [2]. Subsequently, they have been found to be implicated in over thirty diseases [3], notably degenerative diseases such as Parkinson’s or Type-II diabetes. This has prompted much work investigating the structure and formation of amyloid fibrils, as a focus for developing treatments for these diseases [4]. Alongside pathogenic fibrils, amyloid fibrils have been found to fulfil a number of functional roles [5], including mediating surface adhesion [6], biofilm structuring [7], and hormone storage [8]. Due to their attractive material and mechanical properties, amyloid fibrils have also been investigated as components in synthetic systems [5,9], with applications in areas including biomaterials [10] and optoelectronics [11]. Understanding the formation of fibrils, both pathogenic and functional, and how this depends on the environment is therefore of great interest

Commonly it is recognised that the formation of fibrils (and protein aggregation and assembly in general) is affected by surfaces [12,13] and nanoparticles [14,15], which is often attributed to two discrete effects [16]. The first is that, due to their amphiphilic nature, the concentration of proteins near a surface is typically higher than in a bulk solution. This increase in protein concentration then increases the likelihood of proteins aggregating into supramolecular structures, such as fibrils [17]. The second effect comes from the change in the protein conformation on surfaces, due to protein–surface interactions. This can then lead to differences in protein–protein interactions on surfaces, with consequent effects on their aggregation [18,19]. While the first effect increases the rate of fibril formation, the second can promote or inhibit the fibril formation process depending on both protein and surface chemistry. In particular, while hydrophobic surfaces were observed to enhance fibrillation for amyloid beta [20], they inhibited fibril growth for amylin [21]. In the case of amylin molecular dynamics simulations have shown that on hydrophobic surfaces [22] it preferentially adopts alpha-helical structures that may be unfavourable towards fibril formation. This suggests that there is a decoupling between the protein concentration and fibrillation rate, with the relative strengths of the protein–surface and protein–protein interactions playing a critical role [23,24].

Despite the long interest in fibril formation, both in bulk solution and on surfaces, there are many unanswered questions in this. In particular, aspects of the early stages of fibril formation are unclear [25], such as the structure of prefibrillar oligomers. The structure of these can determine the rate of fibril growth as well as influence the structure of the final fibril, which may give rise to fibril polymorphism [26]. It is also thought that the toxic species in some amyloid diseases are off-pathway oligomers [27,28], so understanding oligomer structure and formation may give insight into the progress of degenerative diseases and guide the development of therapies [29]. Due to their small size and relatively transient nature, determining the structure of these small oligomers and their aggregation into fibrils is challenging experimentally [30].

As it operates directly on the molecular level, molecular dynamics simulations have emerged as a powerful tool for the investigation of protein–surface interactions [31,32]. These have been used for a number of years to investigate the adsorption of proteins onto surfaces, revealing molecular details that drive adsorption onto surfaces, such as the role of bound water and surface chemistry and structure. Recently, a number of studies have used these to investigate the behaviour of fibril forming proteins on surfaces [22,33,34,35,36,37,38,39,40], often in combination with advanced sampling techniques, such as replica exchange molecular dynamics [41] or metadynamics [42]. These have shown that changes to surface chemistry can have a significant effect on the conformations the protein can adopt. The specific effect is, however, protein-dependent so it is challenging to draw general conclusions from this.

Simulations have also been used to investigate the formation of protein fibrils [43,44], in particular investigating the early stages of this. These have shown that initially peptides can aggregate into disordered oligomers, with conformational change into beta-strand rich oligomers following this [45,46,47]. While these simulations have largely addressed oligomers in bulk solution, a few simulations have investigated the effect of surfaces on the formation and structure of protein oligomers [35,48,49,50,51]. Notably the amyloidogenic fragments GNNQQNY (from SUP35 prion protein) and NNFGAIL (from amylin) are both found to form fibrillar structures on gold surface [35], mediated by attachment onto already adsorbed peptides. This was in agreement with experimental work that showed that gold nanoparticles enhanced fibril formation for these peptides [52]. Simulations have also addressed the adsorption of fibrils onto surfaces, examining the fibril–surface interaction and changes to fibril structure for different surface chemistries [48,49].

In this paper, the structures formed by aggregates of the amyloidogenic peptide Aβ(16–22) (KLVFFAE) on the Au111 surface and in bulk solution are investigated using molecular dynamics simulations. This is a key fibril forming segment from the amyloid-beta protein [53] and due to its relatively small size, it has attracted significant attention. Single-molecule simulations have also shown that, unlike the GNNQQNY and NNFGAIL sequences studied previously, on gold surfaces this does not adopt fibril-like conformations [34,40]. This makes the combination of the Aβ(16–22) peptide and Au111 surface an ideal test case for investigating the interplay between peptide conformation and interpeptide interactions in oligomerization. Gold nanoparticles have also attracted significant interest in medical applications [54], so understanding the interaction between gold surfaces, for which the Au111 surface is the most thermodynamically stable, is of relevance to medicine. Unlike previous studies of peptide aggregation on surfaces, we are interested in isolating the specific roles of surfaces and interpeptide interactions on this. In order to specifically investigate the role of surfaces on the aggregation of Aβ(16–22) simulations of aggregation of initially isolated monomers on the surfaces will be compared to aggregation in bulk solution. This allows us to examine whether the peptides retain the fibril unfavourable conformation adopted by individual peptides on surfaces during the aggregation process or whether the interpeptide interactions drive the formation of fibril-like conformations. In addition, the adsorption of oligomers formed in solution onto the Au111 surface was investigated to determine the change in oligomer structure due to interaction with the surface.

## 2. Model and Methodology

For both surface and solution simulations, the systems contained four to eight Aβ(16–22) (KLVFFAE) molecules. The initial conformations of the peptides were taken to be the most common conformations found from previous simulations of single Aβ(16–22) peptides in solution and on the Au111 surface [40] (Figure 1). The termini and polarizable residues (K and E) were charged, as appropriate for pH 7. As the peptide is overall charge neutral no counter-ions were needed to neutralise the system. The initial conformations were taken to be the most common structures found from previous simulations of single Aβ(16–22) peptides in solution and the Au111 surface [40] (Figure 1). For bulk simulations, the peptides were randomly placed in the box using the packmol program [55]. These were then solvated through the addition of water molecules, removing any water molecules that overlap with the peptides. The peptide concentrations were 0.05–0.10 mol L${}^{-1}$; this is considerably higher than the peptide concentrations used in the experiment but allowed for the formation of oligomers within the simulation timescales.

In the surface simulations the surface consisted of a 20 × 12 unit cell Au111 surface, with five layers of atoms. A lattice parameter of 4.14 Å was used, consistent with previous work [56]. Positions of the surface atoms, except for the mobile charge sites (see below), were held fixed in the simulations. For simulations investigating aggregation on the surface the peptides were placed in a regular lattice on the surface with random orientations. The initial peptide structure was taken to be the most commonly found conformation from simulations of single Aβ(16–22) peptides on the Au111 surface, with the molecule oriented so the Phe residues were facing the surface. The surface dimensions were 5.860 nm × 6.090 nm, giving a surface density of 1.86–3.92 × 10${}^{-7}$ mol m${}^{-2}$. In the oligomer adsorption simulations, the final oligomer structure from each of the bulk solution simulations (with all the peptides in the system in a single oligomer) was used. These were oriented with their long axis aligned along the *x*-axis and placed so the closest distance between any atom and the gold surface was 10 Å.

All simulations were run in duplicate from different starting configurations. For all cases, the initial structure was energy minimised, with a tolerance of 10${}^{3}$ kJ mol${}^{-1}$ nm${}^{-1}$. For the surface simulations, the *z*-box length was adjusted so the water density far from the surface was equal to the bulk water density, with this typically being ∼7.7 nm.

The charmm22* force field [57,58,59] was used for modelling the peptides with the charmm-TIP3P model [60] used for water. The gold surface was modelled using the polarizable GolP-Charmm model [56,61]. This introduces a mobile charge attached to each gold atom to mimic the effect of polarization and additional interstitial interaction sites on the surface to ensure favourable adsorption on top of the surface gold atoms.

All simulations were run at 300 K, with the solution simulations performed at 1 atm. The temperature was controlled using a velocity-rescale thermostat [62], with a relaxation time of 0.2 ps, while the pressure in the solution simulations was controlled using the Parinello-Rahman barostat [63] (with a relaxation time of 2 ps). Long-range electrostatic interactions were accounted for using a particle mesh Ewald (PME) sum [64], with a real space cutoff of 11 Å. For the solution simulations, 36 wavevectors were used in each direction, for the surface simulations there were 40 wavevectors in the *x* and *y* directions and 52 in the *z*-direction. Short-range van der Waals interactions were cut-off at 11 Å. The simulation timestep was 2 fs, with bonds involving hydrogen atoms constrained using the LINCS [65] and water molecule geometries constrained using the SETTLE algorithms. All simulations were run for 500 ns, with (unless otherwise stated) averages calculated over the final 100 ns. This time was sufficient for the distribution of oligomer sizes to become approximately constant (Figure A1).

The simulations were performed using the Gromacs MD package (version 2018.4) [66,67], with analysis of the simulations performed using standard gromacs utilities and in-house python scripts written using the MDAnalysis library [68]. Simulation snapshots were generated using Visual Molecular Dynamics (VMD) [69]. Peptide oligomerization was monitored using a cluster analysis; two peptides were considered to be part of the same cluster if the distance between two heavy (non-hydrogen) atoms was less than 4 Å. This condition was also used for the calculation of the residue-reside contact maps. The similarity of the peptide structures to reference structures was monitored through the distance RMSD (DRMSD), calculated according to
(1)$$DRMSD=\sqrt{\frac{1}{N_{C\alpha -C\alpha }}\sum_{i}{\left(r_{C\alpha -C\alpha ,i}^{oligomer}-r_{C\alpha -C\alpha ,i}^{ref}\right)}^{2}}.$$

Here, the sum runs over all the Cα−Cα pairs in the peptide. In this work, this was calculated for two reference structures; for the structure of Aβ(16–22) in an amyloid fibril ($DRMSD_{f}$), structure taken from pdb entry 2y29 [70] and $DRMSD_{s}$ from previous simulations of a single Aβ(16–22) peptide on the Au111 surface [40].

## 3. Results

### 3.1. Oligomerization of Aβ(16–22) on Au111 Surfaces and in Solution

In both the surface and solution simulations rapid aggregation is seen for the different systems (Figure 2a). This is unsurprising due to the high concentration of molecules in the simulation. Across the different simulations aggregation within 50 ns is seen. The aggregation is more rapid for the Nmol = 8 system for both surface and solution simulations, consistent with decreased lag time seen for higher protein concentrations [71]. However, as the concentration in all systems is significantly higher than in experimental systems no quantitative conclusions will be drawn from this. While some variation is seen, this typically contains all the peptides, indicating relatively strong binding of the peptides. As the number of peptides in the system increases, the fluctuations in the size of the largest oligomer tend to decrease, suggesting stronger binding of molecules into these larger aggregates.

The differences in stability of the different aggregates can be seen by considering the probability of different oligomer sizes seen in the simulations (Figure 2b). As suggested by the time variation of oligomer size, the probability of finding all the peptides in a single aggregate typically increases with the total number of peptides. For the surface simulations, the larger aggregates are generally more likely than in solution, suggesting stronger aggregation on surfaces. Indeed for the eight peptide simulations, only a single oligomer containing all the peptides is seen for the surface simulations. Variations between different simulation runs for the same number of peptides are seen; notably for six peptides one of the surface simulations has a much lower probability of forming larger oligomers.

The rapid oligomerization can be seen visually through simulation snapshots. Shown in Figure 3 are representative snapshots showing the evolution of the oligomers across the first Nmol = 4 and Nmol = 8 simulations for the Au111 surface. While at the beginning of the simulation, these are placed on a regular grid and not in contact with each other, within the first 10 ns these have typically aggregated into a single oligomer. For Nmol = 4, this is relatively weakly bound and goes through a number of structural arrangements across the simulation. The structure of the larger Nmol = 8 aggregate remains more stable throughout the simulation, consistent with the oligomer size histogram (Figure 2b). Note that for Nmol = 8 the aggregate stretches across the entire simulation box.

In the bulk solution, the initial aggregation is somewhat slower and the initial oligomers are less strongly bound than on the Au111 surface. After about 50 ns disordered amorphous oligomers are formed (Figure 4). Across the simulations, they restructure into more compact and ordered oligomers, with peptides adopting beta-strand-rich structures. This two-step process of fibril formation has been observed in previous studies of aggregation of amyloidogenic peptides.

### 3.2. Surfaces Drive Formation of Linear Aggregates

The tendency towards roughly linear structures on the Au111 surface, particularly for larger Nmol, is seen for all the simulations (Figure 5). As seen in previous simulations the phenylalanine residues lie flat against the gold surface [34,40]. These are typically at the centre of the oligomer, leading to many contacts between these. Typically the charged, terminal residues (lysine and glutamic acid) are found on the outside of the aggregates.

The tendency towards the formation of linear, monolayer structures can be quantified through the eigenvalues of the oligomer gyration tensor (Figure 5b). $G_{min}$ is approximately the same for all the simulations and is comparable to that of an individual peptide molecule. This suggests that the strong adsorption of Aβ(16–22) onto the Au111 surface leads to monolayer formation. Similar to $G_{min}$, $G_{mid}$ is also typically similar for different Nmol. This is comparable to twice that of a single molecule, consistent with the formation of linear structures two molecules wide. An exception to this is found in the first Nmol = 8 simulation, where zig-zag structures are formed. The largest eigenvalue increases with Nmol consistent with the typically linear structures formed.

The structures formed on the Au111 surface are associated with specific interactions between residues, as shown by the intermolecular residue–residue contact maps (Figure 5c). In all cases, contacts are predominately formed by the two phenylalanine residues. In particular, for larger Nmol these form the centre of the linear structures. Contacts between the two charged terminal residues (lysine and glutamic acid) are also found.

A major driving force for the oligomer structure on the gold surface is the strong adhesion of the phenylalanine residues to the gold surface [72,73]. This can be seen in the average residue center-of-mass *z* coordinates (Figure 6). In almost all cases, the two phenylalanine residues are in contact with the surface, consistent with the single molecule structure [34,40]. This increases the probability of contact between phenylalanine residues in different peptides. Across all the surface simulations, only one molecule is found without both phenylalanine residues in contact with the surface. The other residues are typically further from the surface, with the lysine and glutamic acid residues typically the furthest. This is consistent with the strong interaction between phenylalanine residues and the Au111 surface; previous calculations have found the adsorption-free energy of phenylalanine on Au111 ∼−20 kcal mol${}^{-1}$ [72]. Other simulations have found strong attraction between gold surfaces and other aromatic residues such as tyrosine [35]. The interaction with the surface is weaker for the other residues and so these are typically found further from the surface. In particular, the lysine and glutamic acid residues are far from the surface, due to their highly hydrophilic nature, which increases their freedom to move and so increases the probability of contact between these (Figure 5c).

Compared to the Au111 surface in bulk solution Aβ(16–22) forms compact aggregates (Figure 7a). In most cases, these are beta-strand-rich. The gyration tensor eigenvalues (Figure 7b) suggest these are typically elongated along one direction. The shorter two axes are comparable to the length of a single molecule, suggesting that the molecules lie normal to the long axis, similar to an amyloid fibril.

In comparison to the surface simulations, more contacts are found for peptide aggregates in solution (Figure 7c). These typically involve the hydrophobic core (LVFF) of the peptide, implying that this is driven primarily by the hydrophobic effect. The contact maps suggest that the peptides adopt anti-parallel alignment, consistent with the packing seen in fibrils formed by this peptide [70,74]. As for the surface simulations contacts between the termini due to the electrostatic attraction are found.

The different structures can also be characterised by the interpeptide hydrogen bonds formed (Figure 8). Significant differences between these are found between the aggregates formed on the surface and in solution. In most cases, only a small number of interpeptide hydrogen bonds are found for aggregates formed on the Au111 surface. More interpeptide hydrogen bonds are found in the bulk solution. Consistent with the contact maps (Figure 7b), these suggest antiparallel alignment of the molecules.

### 3.3. Oligomerization of Aβ(16–22) on Surfaces Does Not Favour Formation of Fibril-Like Conformations

The simulation snapshots suggest that peptides in oligomers formed on surfaces and in solutions adopt very different conformations, with the peptides in solution oligomers adopting beta-strand-rich structures, similar to those in fibrils. To quantify this histograms of the DRMSD (Equation (Equation 1)) relative to fibril and surface conformations are shown in Figure 9. Generally, surface simulations have peaks in the $P(DRMSD_{f})$ at higher values than solution simulations, with $P(DRMSD_{f})$ being 0 for $DRMSD_{f}<1$ Å for almost all surface simulations. This shows that the Au111 surface favours conformations that are unlike those of fibrils, consistent with simulations of single peptides [34,40] The exception to this is for the first Nmol = 6 simulation, where one of the molecules has adopted a *trans*–conformation with only one phenylalanine residue in contact with the surface (Figure 6), which is more similar to the fibril conformation. The higher similarity between the peptide structure in solution and the fibril structure is consistent with the β-strand rich structure seen in these simulations (Figure 7a), with the similarity to the fibril structure depending on the amount of β-strand seen in the simulations.

By contrast, the peak in $P(DRMSD_{s})$ is found at low values for all the surface simulations, showing that, consistent with the simulation snapshots, the peptides adopt structures similar to the conformation of a single molecule on the Au111 surface. This shows that the inter-peptide interactions are not sufficiently strong to overcome the strong attraction between the phenylalanine residues and the surface. For the solution simulations, peptides typically adopt conformations that are unlike the surface conformation. As Nmol increases the probability of finding surface-like increases slightly, these are typically those found on the exterior of the aggregate.

The specific differences between peptide conformations on surfaces and in solution can be seen by considering the backbone conformation in these environments. Shown in Figure 10 are Ramachandran plots from the different simulations. In the surface simulations, these are generally similar regardless of the number of molecules, suggesting that the peptide conformation is largely independent of oligomer size. While the largest population is found in the upper left quadrant, there is a significant population in the lower left quadrant, near the alpha-helix region. By contrast for most of the solution simulations, only the upper left quadrant is heavily occupied, consistent with a more beta-strand-rich conformation than the surface simulations. The exception to this is the second Nmol = 8 simulation where a disordered aggregate was found. For this case, a Ramachandran plot closer to that of the surface simulations is found.

### 3.4. Difference between Oligomers Formed on Surface and Adsorbed from Solution

As Aβ(16–22) aggregates rapidly in solution, it suggests that the adsorption of small aggregates onto surfaces can occur. Due to the differences in the structure of peptides in these compared to the monomer structure on the surface, this could lead to different oligomer structures as well as differences in peptide conformation within these. Starting from aggregates placed in the vicinity but not in contact with the surface in all cases these adsorb rapidly, indicating that in common with single peptide molecules there is a strong interaction with the surface. Note that during this initial adsorption, the peptides can detach from the oligomers, suggesting that the interaction with the surface is of comparable strength to the peptide–peptide interactions.

While rapid adsorption is seen in cases, the shape of the aggregate differs between the simulations. Notably, both monolayers, where all the peptides are in contact with the surface, and multilayer structures are found (Figure 11b). The adoption of different oligomer structures by amyloidogenic molecules on surfaces has been seen previously for amyloid beta oligomers on different self-assembled monolayers [49].

The change in aggregate shape can be quantified through the gyration tensor eigenvalues (Figure 12a). Notably, for simulations in which a monolayer forms $G_{min}$ is similar in value to that of the oligomers formed on surfaces (Figure 5b), while it is larger when a multilayer aggregate is found. $G_{max}$ and $G_{mid}$ are typically larger then those for the solution aggregates. This change, however, is not always true with the oligomer structure in some cases (e.g., run2 for Nmol = 4) remaining similar to those for the solution simulation. The largest difference between the oligomer structure is found for Nmol = 8, run 2; this case corresponds to an oligomer without any beta-strand content, which may lead to weaker interpeptide interactions within the oligomer, allowing for greater changes in structure.

The differences in oligomer structure are also reflected in changes to the contacts between residues in the aggregates (Figure 12b). Simulations in which multilayer aggregates are formed are similar to those for aggregates in bulk solution, with many contacts between the hydrophobic core of the peptides. Fewer contacts are typically found for the monolayer aggregates, with contacts involving the phenylalanine residues being more prominent; this is similar to the contact maps for aggregates formed on the surface. In all cases, contacts between the charged termini are found, similar to the other oligomers.

Changes in the aggregate shape are reflected in the positions of individual residues with respect to the surface (Figure 13). As for the simulations starting from isolated peptides on the surface commonly phenylalanine residues are found near the surface. In a number of cases, both the phenylalanine residues for a particular peptide are in contact with the surface, suggesting that these adopt surface-like structures.

Changes to the single peptide conformation are also seen during the adsorption of the oligomer. Compared to bulk solution, there is a decrease in the probability of finding fibril-like structures and more surface-like conformations are found (Figure 14). This shift is largely due to the molecules lying closest to the surface, with the structural change driven by the strong interaction between the surface and phenylalanine residues.

## 4. Conclusions

Surfaces and interfaces have long been known to affect the aggregation and assembly of proteins and other biomolecules, with the formation of amyloid fibrils being a particularly important and interesting case. While the protein concentration on a surface is typically higher than in the bulk solution, which would naturally increase the aggregation rate, surfaces can both promote or inhibit the formation of fibrils. This suggests that the fibrillation process depends on details of the protein conformation on the surface and on the protein–surface interaction. To gain insight into this, in this paper molecular dynamics simulations were used to investigate the aggregation of the model amyloidogenic peptide, Aβ(16–22), on gold surfaces and in bulk solution.

While aggregation on simulation timescales is seen in both surface and solution simulations, the structures of aggregates that were formed differed markedly. In particular, while the solution aggregates were typically rich in beta-strands, these were not found in the aggregates formed on surfaces. This suggests that despite the more rapid aggregation of peptides on the surface, fibril formation may be inhibited, consistent with experiment [34]. On the surface the peptide conformation in the oligomers is similar to that of isolated peptides [40], with the likelihood of them adopting fibril-like conformations small. The adoption of non-fibril-like conformations on surfaces may be a common feature of nanomaterial systems that can inhibit the formation of fibrils [75]. These differences in structure are driven both by the effect of the surface on peptide conformation and by the different interactions found between peptides. On the surface, the most common interactions are between phenylalanine residues, which are pinned to the surface, and between the charged termini. In solution, aggregation is largely driven by hydrophobic interactions [76]. Note that in the present study, the end termini are uncapped (as in previous simulations of the single peptide system [40]). As the two terminal residues in this peptide have charged side chains at neutral pH, charge–charge interactions would still be present if the peptide ends were capped, so qualitatively similar results may be expected in that case. Nonetheless, different end capping groups have been shown to affect the behaviour of peptides on surfaces [77] and this would be an interesting avenue for future work.

While there is some variation in the stability of the different sizes aggregates formed on the Au111 surface (Figure 2b), both the oligomer structures (Figure 5) and single peptide conformations (Figure 10) were similar for the different numbers of peptides. This suggests that the tendency of the surface to disfavour fibril formation holds regardless of the system size.

Aggregation in solution means that as well as adsorbing individually peptide aggregates can adsorb onto the surface. Simulations of aggregates show that these may also rapidly adsorb onto the surface. This is accompanied by changes in the structure of the aggregate and individual peptides. In some cases, they adopt monolayer structures with every peptide in contact with the surface, while in others multilayers are formed. This does not depend on the number of molecules in the aggregate. The structures of the peptides in the aggregates also change, with those in contact with the surface being more liable to adopt surface-like conformations.

As with all simulations the chosen starting conditions can affect the final results. From previous simulations of single peptides on the Au111 surface, the two phenylalanine residues are almost always found in contact with the surface so the conformation is likely to be a reasonable starting structure for the peptide. The weaker adsorption of the other residues onto the surfaces means that changes from the initial structure are seen in the simulation timescales as suggested by the histograms of the $DRMSD_{s}$ (Figure 9). The strong attraction for the phenylalanine residues for the surface is also seen in the oligomer adsorption simulations, where adsorption is through the phenylalanine residues and change of peptide structure to more surface-like conformations is seen.

While these simulations are consistent with the effect of gold surfaces on the fibrillation of Aβ(16–22), as with all molecular dynamics studies the time and length scales that have been investigated are limited. In particular, the concentrations are far above those that would be found in typical experimental systems. The timescales of the simulations were also too short to see if aggregates adsorbed from solution onto surfaces would always completely transform to surface-like aggregates. To investigate this coarse-grained models [78] can be used to extend the accessible time scales. Markov state models [79] or kinetic Monte Carlo [80] models could also be used to give more quantitative information on the aggregation kinetics.

## Figures and Tables

**Figure 1 biomolecules-13-01261-f001:**
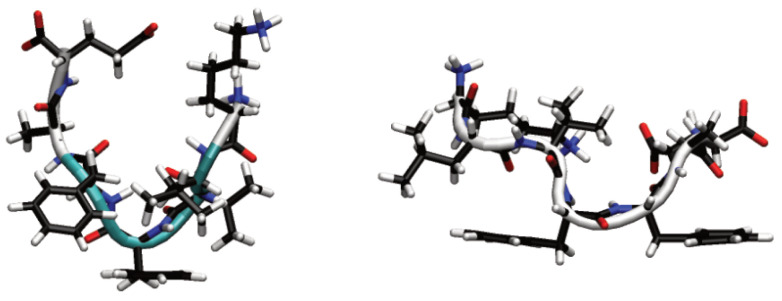
Initial peptide conformations for solution (**left**) and Au111 surface (**right**). Backbone secondary structure shown in cartoon form, with white denoting coil and turquoise turn.

**Figure 2 biomolecules-13-01261-f002:**
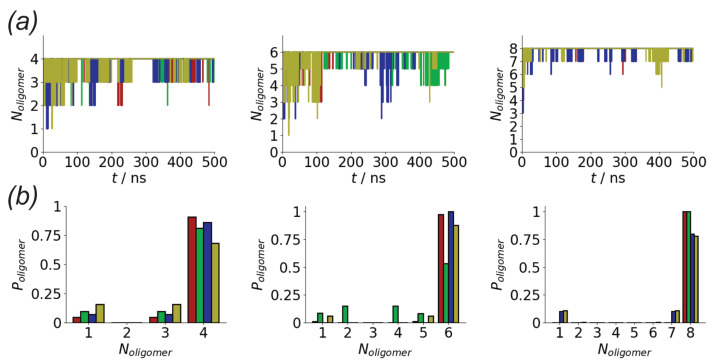
(**a**) Variation in largest oligomer size for Nmol = 4 (**left**), Nmol = 6 (**middle**), and Nmol = 8 (**right**) simulations. Red and green denote surface simulations and blue and gold denote solution simulations. (**b**) Histograms of oligomer sizes for Nmol = 4 (**left**), Nmol = 6 (**middle**), and. Nmol = 8 (**right**) simulations, calculated over the last 100 ns of each simulation. Colours as in (**a**).

**Figure 3 biomolecules-13-01261-f003:**
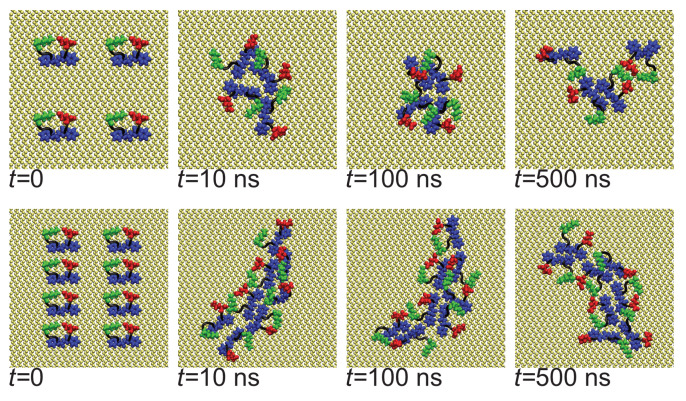
Snapshots showing evolution of oligomer structure of Aβ(16–22) on Au111 surface. (**Top**) panel shows Nmol = 4, (**bottom**) Nmol = 8. In all snapshots phenylalanine, glutamic acid, and lysine residues highlighted in blue, red, and green, respectively.

**Figure 4 biomolecules-13-01261-f004:**
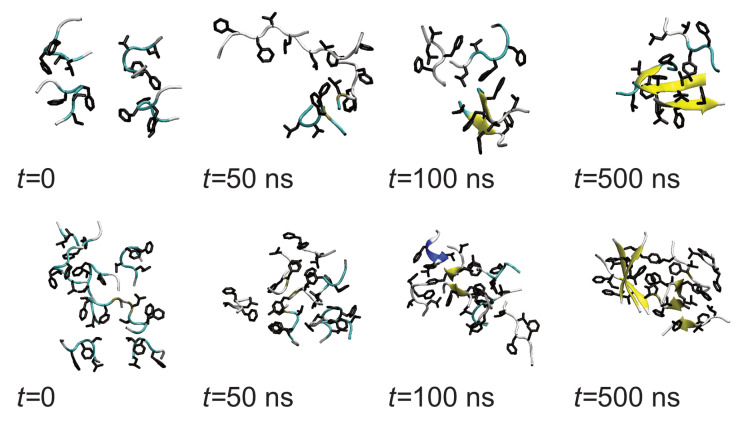
Snapshots showing evolution of oligomer structure of Aβ(16–22) in bulk solution. (**Top**) panel shows Nmol = 4, (**bottom**) Nmol = 8. In all snapshots hydrophobic residues are highlighted.

**Figure 5 biomolecules-13-01261-f005:**
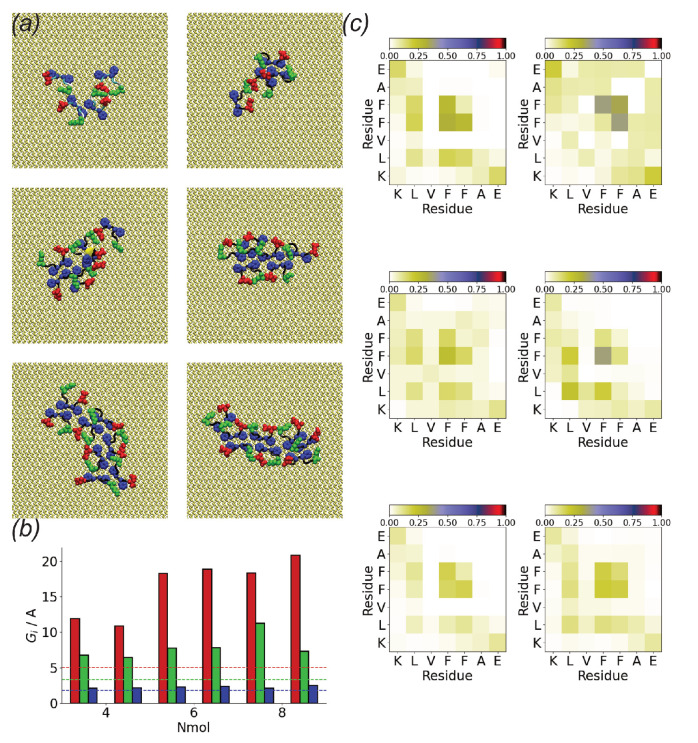
(**a**) End simulation snapshots for Aβ(16–22) on Au111 surface for Nmol = 4 (**top**), Nmol = 6 (**middle**), and Nmol = 8 (**bottom**). In all cases, phenylalanine, glutamic acid, and lysine residues are highlighted in blue, reg, and green, respectively; (**b**) Oligomer gyration tensor eigenvalues for Aβ(16–22) on Au111 surface. Red, green, and blue denote $G_{max}$, $G_{mid}$, and $G_{min}$, respectively. Dashed lines show gyration tensor eigenvalues for single molecules; (**c**) Residue-residue contact maps for Nmol = 4 (**top**), Nmo l= 6 (**middle**), and Nmol = 8 (**bottom**) for Aβ(16–22) on Au111 surface.

**Figure 6 biomolecules-13-01261-f006:**
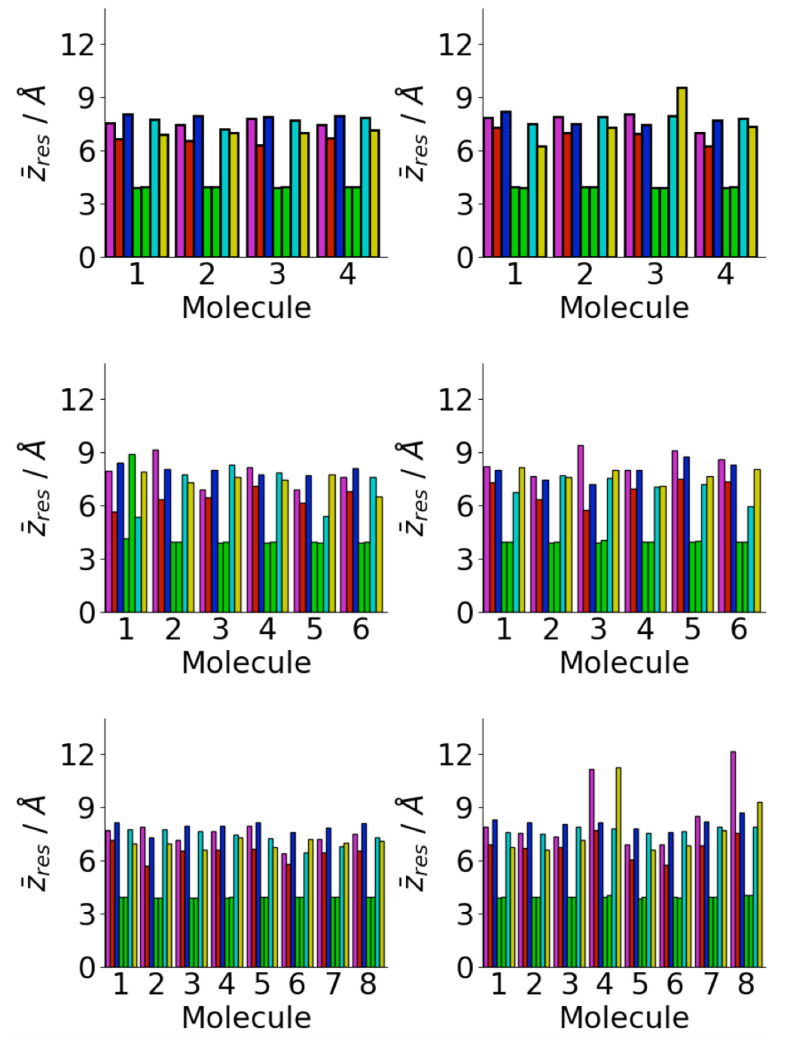
Residue centre-of-mass average *z* coordinates for Aβ(16–22) on Au111 surface for Nmol = 4 (**top**), Nmol = 6 (**middle**), and Nmol = 8 (**right**). Magenta, red, blue, green, cyan, and gold denote lysine, leucine, valine, phenylalanine, alanine, and glutamic acid residues, respectively.

**Figure 7 biomolecules-13-01261-f007:**
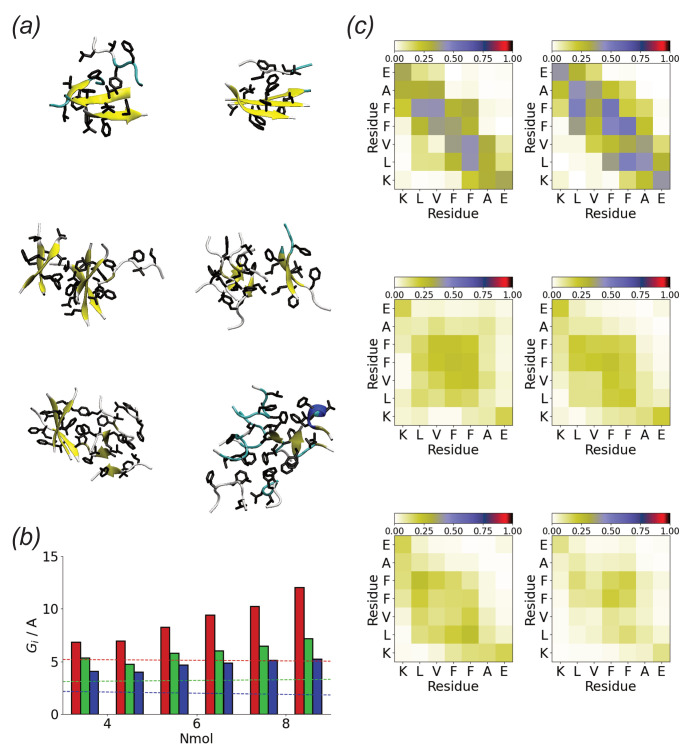
(**a**) End simulation snapshots for Aβ(16–22) in solution for Nmol = 4 (**top**), Nmol = 6 (**middle**), and Nmol = 8 (**bottom**). In all cases, the peptide backbone represented in the cartoon and hydrophobic residues are highlighted; (**b**) Oligomer gyration tensor eigenvalues for Aβ(16–22) in solution. Red, green, and blue denote $G_{max}$, $G_{mid}$, and $G_{min}$, respectively. Dashed lines show gyration tensor eigenvalues for single molecules; (**c**) Residue-residue contact maps for Nmol = 4 (**top**), Nmol = 6 (**middle**), and Nmol = 8 (**bottom**) for Aβ(16–22) in solution.

**Figure 8 biomolecules-13-01261-f008:**
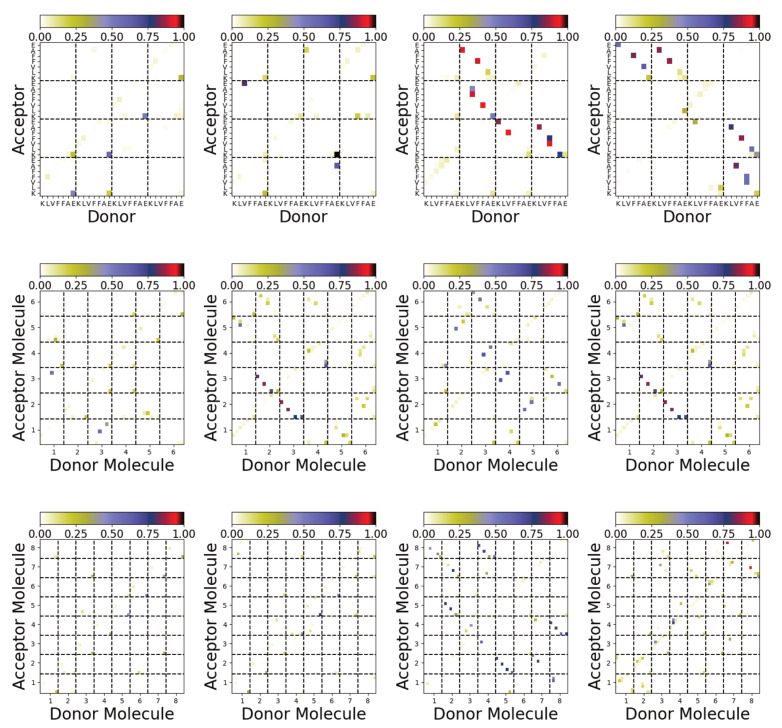
Hydrogen bonding maps from simulations of Aβ(16–22) for Nmol = 4 (**top**), Nmol = 6 (**middle**), and Nmol = 8 (**bottom**). The leftmost two columns are simulations on the Au111 surface and the rightmost two columns are from simulations in bulk solution.

**Figure 9 biomolecules-13-01261-f009:**
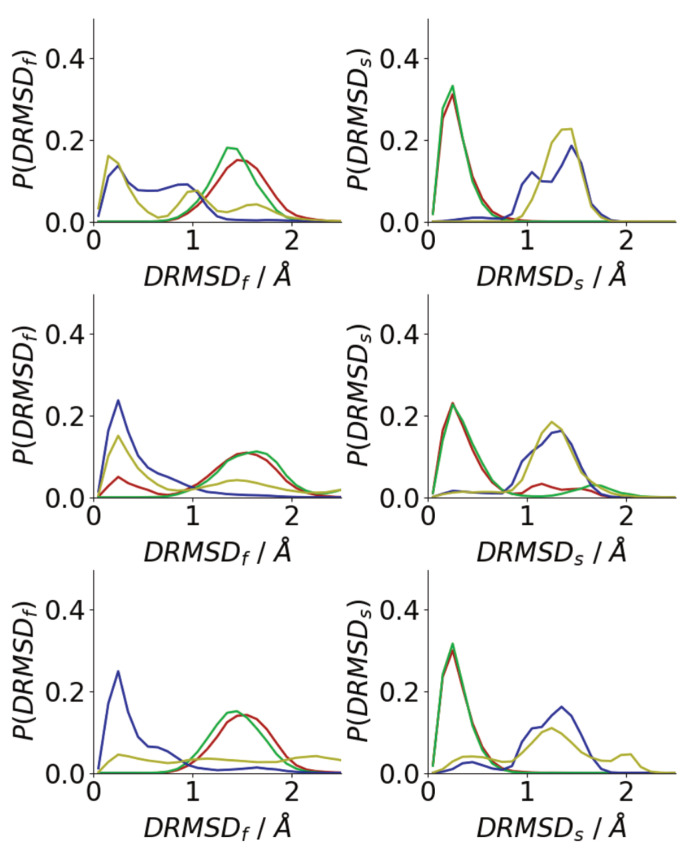
Histograms of DRMSD between fibril (**left**) and surface (**right**) for Nmol = 4 (**top**), Nmol = 6 (**middle**), and Nmol = 8 (**bottom**). Red and green denote surface simulations and blue and gold denote solution simulations.

**Figure 10 biomolecules-13-01261-f010:**
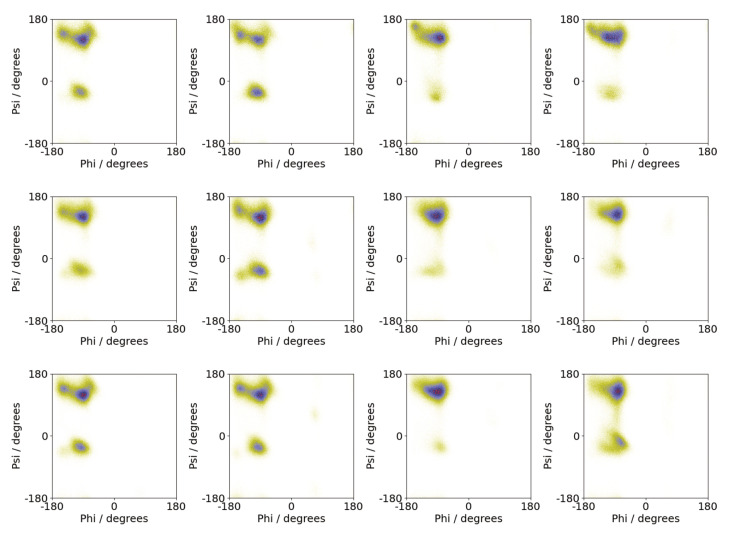
Ramachandran plots from simulations of Aβ(16–22) for Nmol = 4 (**top**), Nmol = 6 (**middle**), and Nmol = 8 (**bottom**). The leftmost two columns are simulations on the Au111 surface and the rightmost two columns are from simulations in bulk solution.

**Figure 11 biomolecules-13-01261-f011:**
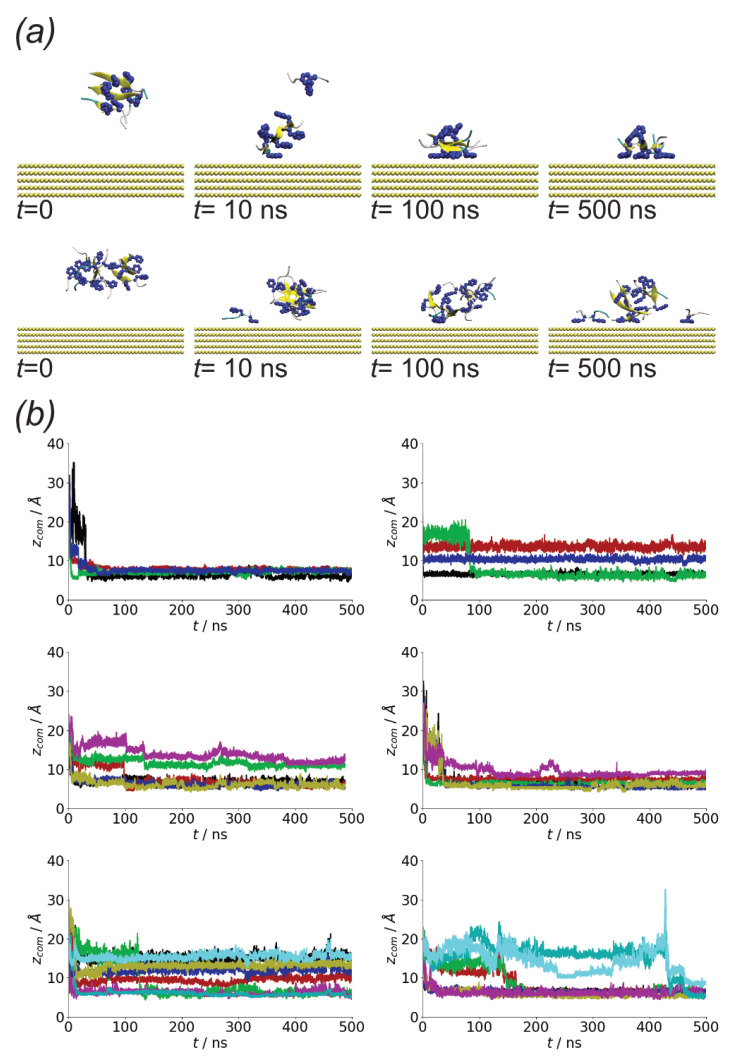
(**a**) Simulation snapshots for oligomer adsorption, taken from Nmol = 4 run1 (**top**) and Nmol = 8, run1 (**bottom**). (**b**) Protein centre-of-mass *z* coordinates from (**top** to **bottom**) Nmol = 4, Nmol = 6, and Nmol = 8. Left and right-hand columns show runs 1 and 2 for each system, respectively.

**Figure 12 biomolecules-13-01261-f012:**
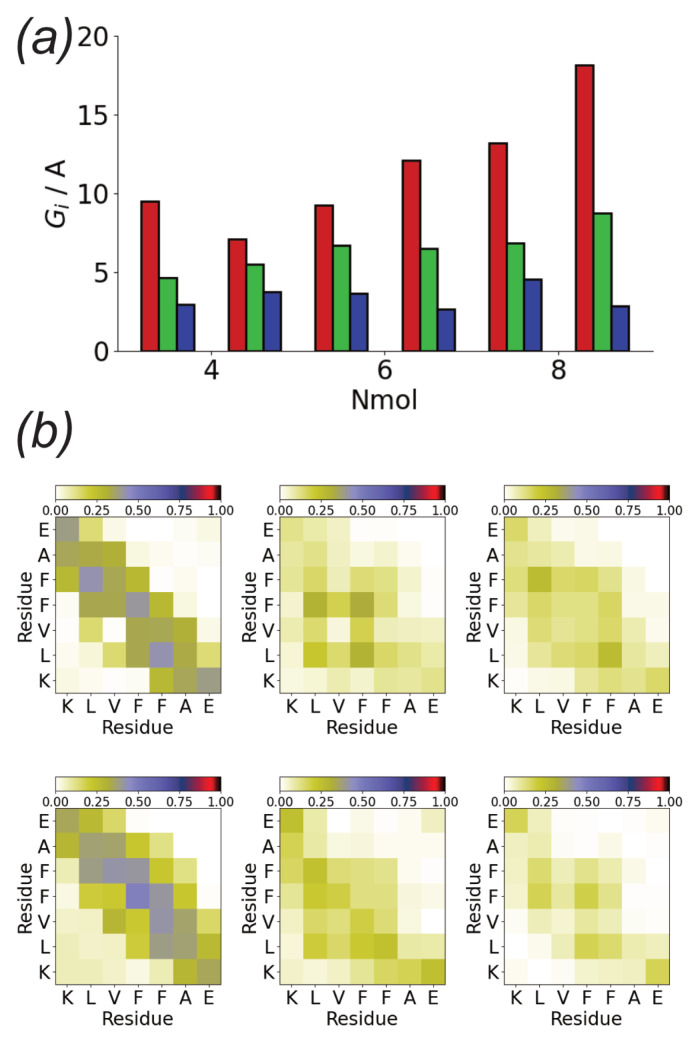
(**a**) Gyration tensor eigenvalues for aggregate adsorption simulations. Red, green, and blue denote $G_{max}$, $G_{mid}$, and $G_{min}$, respectively. (**b**) Contact maps for aggregate adsorption simulations for Nmol = 4 (**top**), Nmol = 6 (**middle**), and Nmol = 8 (**bottom**). Left and right-hand columns show runs 1 and 2 for each system, respectively.

**Figure 13 biomolecules-13-01261-f013:**
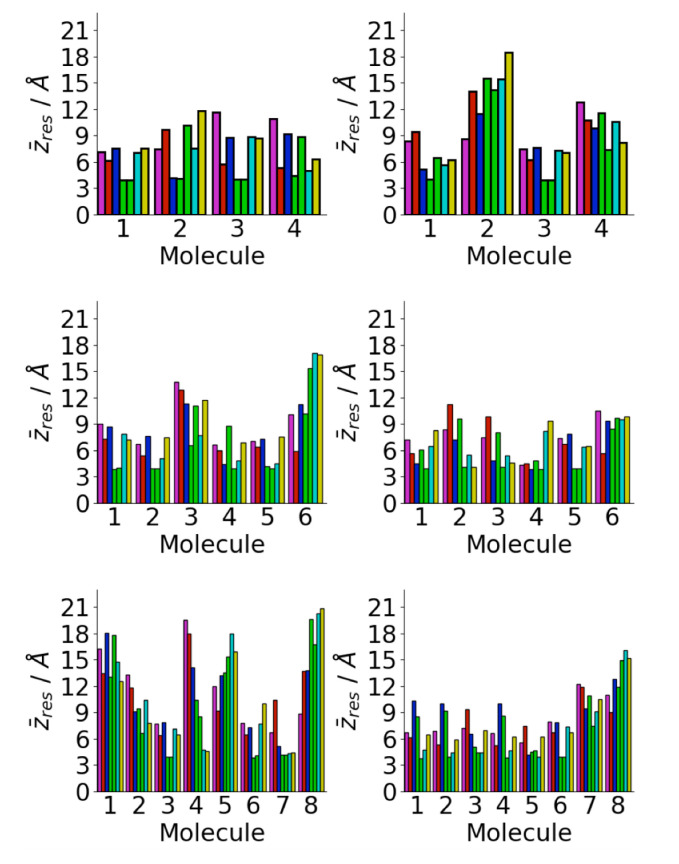
Residue centre-of-mass average *z* coordinates for aggregate adsorption simulations for Aβ(16–22) on Au111 surface for Nmol = 4 (**top**), Nmol = 6 (**middle**), and Nmol = 8 (**right**). Magenta, red, blue, green, cyan, and gold denote lysine, leucine, valine, phenylalanine, alanine, and glutamic acid residues, respectively.

**Figure 14 biomolecules-13-01261-f014:**
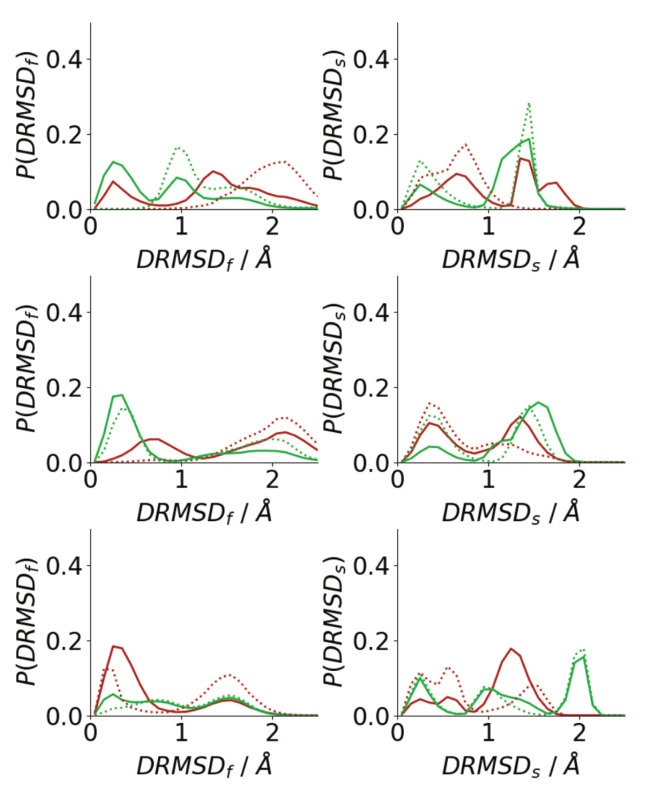
Histograms of DRMSD between fibril (**left**) and surface (**right**) for Nmol = 4 (**top**), Nmol = 6 (**middle**), and Nmol = 8 (**bottom**). Solid lines denote histograms calculated for all peptides, dotted lines those calculated for peptides in contact with the surface.

## Data Availability

The data presented in this study are available in the article.

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
