# Peer review of "Aggregation of an Amyloidogenic Peptide on Gold Surfaces"

_biomolecules, 2023, doi:10.3390/biom13081261_

Round 1
Reviewer 1 Report
In the “Aggregation of an Amyloidogenic Peptide Gold Surfaces”. By using molecular dynamics simulations, Cheung investigates the effect of a Au(111) surface on the conformational and aggregation properties of the Aβ(16-22), from the amyloid beta protein. The author concludes that gold surface reduces the probability of forming fibril-like aggregates. The novelty of this finding is not high, as this evidence has already been reported both experimentally and computationally. Remaining on the computational side, for example, Menziani and co-workers reported similar conclusions (on the 16-22 tract) by using enhanced sampling techniques on the whole Aβ(1–40) tract in the presence of citrates-capped gold nanoparticles; overall, conditions closer to that of interest, with respect to those reported by Cheung.
Cheung also concludes that this result is due to the tendency of phenylalanine residues to interact with the gold surface and of the charged termini to interact between them. Both these results could be affected by the chosen starting conditions. On the one hand, the author places the peptides with the phenylalanine direct toward the gold surface, form the other hand, he uses charged termini, which for sure amplify the attraction between the N (double positively charged) and the C-terminus (double negatively charged). At least in one case, I suggest simulating systems starting from different configurations and/or with neutral termini (a condition closer to reality, where the simulated sequence is part of a larger tract).
Finally, a few author conclusions need to be supported by more quantitative data. What about the content of different secondary structures? Have the author performed a DSSP analysis of the different simulations? What about the Ramachandran plots? And the inter-chains H-bonds? These data could be useful to support the claim that the peptide adopts different conformations in different conditions (e.g., at line 167 or at line 220)
Other minor comments:
1) In figure 1, the effect of the number of monomers on the stability seems to be quite evident. Does the author think that also the tendency to disfavor the fibril conformation can be affected by dimension? Data quantitatively related to the secondary structures in different conditions (see above) could help to discuss this point
2) Again, regarding figure 1: also the time to reach the bigger aggregate seems to decrease in the system with 8 peptides, suggesting a kinetic effect. Are there experimental pieces of evidence for that?
3) Line 106: It needs to be clarified the secondary structure adopted for the starting conformation.
4) Data reported in figure 7 need to be better clarified.
5) Line 88-90: the sentence “Peptide oligomers that adsorb onto the gold surface can undergo rearrangement into monolayer aggregates, with peptide structures similar to those found on surfaces” is not clear
6) At line 66, simulations instead than simulation
Author Response
In the “Aggregation of an Amyloidogenic Peptide Gold Surfaces”. By using molecular dynamics simulations, Cheung investigates the effect of a Au(111) surface on the conformational and aggregation properties of the Aβ(16-22), from the amyloid beta protein. The author concludes that gold surface reduces the probability of forming fibril-like aggregates. The novelty of this finding is not high, as this evidence has already been reported both experimentally and computationally. Remaining on the computational side, for example, Menziani and co-workers reported similar conclusions (on the 16-22 tract) by using enhanced sampling techniques on the whole Aβ(1–40) tract in the presence of citrates-capped gold nanoparticles; overall, conditions closer to that of interest, with respect to those reported by Cheung. While the effect of surfaces on the conformation and aggregation of amyloidogenic peptides and proteins has been the subject of a number of studies (as detained in the introduction) this paper aimed to investigate the specific differences between the aggregation and oligomerization of a model amyloidogenic peptide on surfaces and in bulk solution. The choice of starting states (initially isolated monomers on the surface and in solution) was made to isolate differences between aggregation on surfaces and in solution, which had not been previously done. This is now discussed in the introduction (lines 88-98). Cheung also concludes that this result is due to the tendency of phenylalanine residues to interact with the gold surface and of the charged termini to interact between them. Both these results could be affected by the chosen starting conditions. On the one hand, the author places the peptides with the phenylalanine direct toward the gold surface, form the other hand, he uses charged termini, which for sure amplify the attraction between the N (double positively charged) and the C-terminus (double negatively charged). At least in one case, I suggest simulating systems starting from different configurations and/or with neutral termini (a condition closer to reality, where the simulated sequence is part of a larger tract). As with all simulations the chosen starting conditions can affect the final results. From previous simulations of single peptides on the Au(111) surface the two phenylalanine residues are almost always found in contact with the surface so we believe that this is a reasonable starting structure for the peptide. The weaker adsorption of the other residues onto the surfaces means that changes from the initial structure are seen in the simulation timescales (as suggested by the histograms of the DRMSDs), The strong attraction for the phenylalanine residues for the surface is also seen in the oligomer adsorption simulations, where adsorption is through the phenylalanine residues and change of peptide structure to more surface-like conformations is seen. While it is true that commonly capped peptides are used in experimental systems, the aim of this paper is not on comparison to a specific experimental system but rather a comparision between the oligomerization in different environments and how this may explain the effect of surfaces on fibrillation. Also in this specific case the end termini will still be charged so the interactions between these would likely be qualitatively similar. The effect of terminal capping groups on peptide conformation on surfaces is interesting and may be the subject of our future work, but it is not within the scope of this paper. These points are now discussed in the conclusions (lines 339 to 345 and lines 359 to 367) Finally, a few author conclusions need to be supported by more quantitative data. What about the content of different secondary structures? Have the author performed a DSSP analysis of the different simulations? What about the Ramachandran plots? And the inter-chains H-bonds? These data could be useful to support the claim that the peptide adopts different conformations in different conditions (e.g., at line 167 or at line 220) Ramachandran plots and discussion of these have been added to the paper (Figure 10 and lines 265 to 275). As can be seen significant differences are seen between these on the surface and in bulk solution. Secondary structure analysis was performed; the results (shift towards beta-strands for oligomers in solution, and turn and coil rich conformations on the surface) were consistent with the Ramachandran plots so were omitted for the sake of brievity. Comparison Interpeptide hydrogen bonds have been calculated and discussed in the manuscript (Figure 8 and lines 235-240). Other minor comments: 1) In figure 1, the effect of the number of monomers on the stability seems to be quite evident. Does the author think that also the tendency to disfavor the fibril conformation can be affected by dimension? Data quantitatively related to the secondary structures in different conditions (see above) could help to discuss this point While the stability of the aggregates increases with the number of monomers, the monomer structures for the surface aggregates remain similar in all cases (as can be seen in the newly added Ramachandran/secondary structure data). Similarly the aggregate structure formed on the Au111 surface is similar for both the Nmol=6 and Nmol=8 systems, suggesting that (at least for the system sizes studied), the tendency to disfavour fibril-like conformations on the surface is not dependent on system size. This is discussed in the conclusions (lines 346-350). 2) Again, regarding figure 1: also the time to reach the bigger aggregate seems to decrease in the system with 8 peptides, suggesting a kinetic effect. Are there experimental pieces of evidence for that? Aggregation tends to occur more rapidly in the denser systems, which is consistent with the decrease in the lag time associated with fibrillation at higher concentrations. This is now discussed in the manuscript (155-159). 3) Line 106: It needs to be clarified the secondary structure adopted for the starting conformation. To clarify the initial structures snapshots showing these have been added (Figure 1) 4) Data reported in figure 7 need to be better clarified. The discussion of Figure 7 (now figure 8) has been amended to make the meaning of this clearer (lines 243-253). 5) Line 88-90: the sentence “Peptide oligomers that adsorb onto the gold surface can undergo rearrangement into monolayer aggregates, with peptide structures similar to those found on surfaces” is not clear Due to revision of final paragraph of the introduction this sentence has been removed. 6) At line 66, simulations instead than simulation This has been corrected.

Reviewer 2 Report
This manuscript investigated the structures formed by aggregates of the amyloidogenic peptide Aβ on the Au111 surface and in bulk solution by using MD. I am not sure what the point of put peptide on Au111? The author need to clarify what this means and compare it with previous studies.
PS. the simulation time 500ns is too short.
no comment
Author Response
This manuscript investigated the structures formed by aggregates of the amyloidogenic peptide Aβ on the Au111 surface and in bulk solution by using MD. I am not sure what the point of put peptide on Au111? The author need to clarify what this means and compare it with previous studies. The aim of the paper is to investigate the role of surfaces in the aggregation of a model amyloidogenic peptide, specfically the amyloid beta(16-22) fragment. As discussed in the introduction interfaces, such as solid surfaces and nanoparticles, have been shown to affect the fibrillation process. The Au111 surface was chosen as previous studies have shown that the peptide adopts signficantly different conformations on this surface than in bulk solution, so it is an ideal test case for investigating the interplay between peptide conformation and interpeptide interactions in oligomerization. Gold nanoparticles have also attracted significant interest in medical applications so understanding the interaction between gold surfaces (for which the 111 facet is the most thermodynamically stable) is of relevance to medicine. This is now discussed in the introduction (lines 83-88). The simulation time 500ns is too short. While the 500 ns time is short on the timescale of experimental studies the aim of this paper was to investigate the early stages of aggregation so this time scale is sufficient. This is shown as in almost all cases features of the system have equilibrated within the simulation time frame. To demonstrate this the oligomer size distributions calculated between for 50 ns intervals across the last 200 ns of the simulation (Figure A1). In most cases there is no systematic variation between these suggesting the the Similarly, as shown in peptide centre-of-mass positions (Figure 9) in almost all cases these have taken on constant values by the end of the simulations, again suggesting that the simulations have reached an equilibrium state.
